# Interaction among Calcium Diet Content, PTH (1-34) Treatment and Balance of Bone Homeostasis in Rat Model: The Trabecular Bone as Keystone

**DOI:** 10.3390/ijms20030753

**Published:** 2019-02-11

**Authors:** Marzia Ferretti, Francesco Cavani, Laura Roli, Marta Checchi, Maria Sara Magarò, Jessika Bertacchini, Carla Palumbo

**Affiliations:** 1Department of Biomedical, Metabolic and Neural Sciences, Section of Human Morphology, University of Modena and Reggio Emilia, 41124 Modena, Italy; marzia.ferretti@unimore.it (M.F.); francesco.cavani@unimore.it (F.C.); marta.checchi@unimore.it (M.C.); mariamariasara.magaro@unimore.it (M.S.M.); 2Department of Laboratory Medicine and Pathological Anatomy, Azienda USL of Modena, 41126 Modena, Italy; l.roli@ausl.mo.it

**Keywords:** calcium diet content, mineral homeostasis, skeletal homeostasis, trabecular bone, PTH (1-34), rat

## Abstract

The present study is the second step (concerning normal diet restoration) of the our previous study (concerning the calcium-free diet) to determine whether normal diet restoration, with/without concomitant PTH (1-34) administration, can influence amounts and deposition sites of the total bone mass. Histomorphometric evaluations and immunohistochemical analysis for Sclerostin expression were conducted on the vertebral bodies and femurs in the rat model. The final goals are (i) to define timing and manners of bone mass changes when calcium is restored to the diet, (ii) to analyze the different involvement of the two bony architectures having different metabolism (i.e., trabecular versus cortical bone), and (iii) to verify the eventual role of PTH (1-34) administration. Results evidenced the greater involvement of the trabecular bone with respect to the cortical bone, in response to different levels of calcium content in the diet, and the effect of PTH, mostly in the recovery of trabecular bony architecture. The main findings emerged from the present study are (i) the importance of the interplay between mineral homeostasis and skeletal homeostasis in modulating and guiding bone’s response to dietary/metabolic alterations and (ii) the evidence that the more involved bony architecture is the trabecular bone, the most susceptible to the dynamical balance of the two homeostases.

## 1. Introduction

Balanced relationships between bone resorption and bone formation in maintaining skeletal health are important during the entire life cycle; in fact, their impairment induces, particularly during aging, bone loss and microarchitectural deterioration, with a consequent increase of bone fragility and susceptibility to fractures [1,2,3,4,5,6,7,8,9,10]. Such balance is obtained by the interaction of manifold factors: lifestyle, general clinical conditions, physical activity, use of osteoprotective drugs, and dietary mineral content. In the case of a calcium-deficient diet, for example, different bone alterations are induced in the various skeletal regions (axial and/or appendicular) having different architecture (trabecular and/or cortical) and metabolism, and that are differently susceptible to loading in the accomplishment of the skeletal homeostasis. Diet is a changing risk factor for bone diseases, such as osteoporosis, and adequate amounts of calcium are essential lifelong to maintain healthy bone mass [11,12]. Among many other nutrients, bone development requires adequate amounts of calcium, which is the most important mineral element, since the skeleton represents the greatest calcium store in the body [13]. In fact, more than 95% of the body’s calcium is present in bone tissue as hydroxyapatite, conferring rigidity, hardness, and structural integrity to the skeleton [14]. Several investigations have underlined the role of serum calcium variations (mainly dependent on dietary calcium uptake) in the etiopathogenesis of osteopenia/osteoporosis and fracture occurrence in both animals and humans [15,16]: low calcemia activates PTH which, in turn (i) stimulates the production in the kidneys of vitamin D, which enhances the calcium uptake in the active sites of rat duodenal mucosa [17]; (ii) decreases the urinary excretion of calcium; and (iii) stimulates calcium resorption from human bone tissue [18]. A fraction of parathyroid hormone, PTH (1-34), is a peptide that regulates calcium and phosphorus metabolism for the mineral homeostasis [19]. Intermittent administration of PTH (1-34) has been shown to induce new bone formation and to increase bone mineral density [20,21,22,23,24,25]. Therefore, PTH (1-34) has been widely used for bone regeneration in osteoporotic patients, on the basis of a great number of data from both clinical studies [26,27,28,29] and experimental animal models [30,31,32,33,34,35,36,37,38]; however, the behavior/target of this drug is not fully clarified.

A previous study of ours, performed on rats who were fed calcium-free diet for one month [39], showed how, after the induction of metabolic osteopenia/osteoporosis, both the mineral and skeletal homeostasis influence the sites of bone loss: trabecular and/or cortical bone in axial and/or appendicular skeleton (i.e., lumbar vertebra and femur). Thus, the extreme lack of calcium in the diet does not lead to only one type of bone response. It is to be underlined, in fact, that the various answers recorded in the different bony architectures are due to the different main involvement of each skeletal region in maintaining mineral or skeletal homeostasis, namely different answers can be evoked by a given experimentally-induced biochemical osteopenia/osteoporosis.

The present study, concerning normal diet restoration, is the second experimental step and all data reported here will be considered together with the previous ones [39] to fully understand the progress of recovery. The aim is to determine whether normal diet restoration (after one month of calcium-free feeding), with/without PTH (1-34) administration, can influence amounts and deposition sites of the total bone mass, also depending on skeletal homeostasis. This model is proposed to study the bone alterations during temporary unbalanced calcium metabolism; the final goals are (i) to define timing and manners of bone mass changes when calcium is restored to the diet, (ii) to verify the different involvements of the two bony architectures having different metabolisms (i.e., trabecular versus cortical bone), and (iii) to verify the eventual role of PTH (1-34) administration during bone mass recovery.

## 2. Results

As mentioned above, the results reported here, collected during diet restoration, will be compared with those concerning calcium-free diet already published [39]. Figure 1 shows the overall experimental conditions of the previous and the present studies, to better understand their meaning.

### 2.1. Body Weight

Table 1 reports the mean values of body weights of all groups recorded at the arrival time (TA) in the housing facility, and at the beginning (T0), after four weeks (T1), and after eight weeks (T2) of the experimentation. All rats show a weight increase of ~5% during the acclimation period. At T1, body weight is significantly higher in all rats with respect to T0, as well as at T2 with respect to T1. No significant differences were found among all groups within each trial period considered.

### 2.2. Histology and Histomorphometry

Morphological and histomorphometrical evaluations were performed on transverse sections of both L5 and right femur (mid-diaphyseal and distal metaphyseal levels) of all rats, at times T1 (i.e., after four weeks) and T2 (i.e., after eight weeks).

#### 2.2.1. Vertebral body (L5)

Figure 2 shows transverse vertebral body sections stained with Alizarin Red, used for static histomorphometry. In rats fed a calcium-free diet for four weeks (groups 3 and 4), bone trabeculae appear thinner and rarefied with respect to controls (groups 2 and 9), especially in the posterior portion of the vertebral body. In normal diet restoration (groups 5, 6, 7, and 8), bone trabeculae appear thicker than those present in rats fed a calcium-free diet, but only in the posterior portion of the vertebral body, still rarefied (even absent).

Figure 3 displays the anterolateral portion of L5 bodies, showing bone labels used for dynamic histomorphometric evaluation.

Histograms reported in Figure 4 and Figure 5 refer to all static and dynamic histomorphometric parameters of trabecular and cortical bone, respectively. In rats fed a calcium-free diet for 4 weeks and successive normal diet restoration for four weeks (groups 6, 7, and 8), independently from PTH administration (during calcium-free diet and/or normal diet restoration), the BV/TV increases with respect to both groups 3 and 4 (fed calcium-free diet for one month) and group 5 (normal diet restoration without PTH administration), notwithstanding the absence of statistical significance; moreover, the BV/TV value (groups 6, 7, and 8) approaches the value recorded in control group 9 (Figure 4). The trabecular thickness (Tb.Th) increases in animals fed with a normal diet restoration with respect to animals fed a calcium-free diet, with statistical significance, mainly in rats that underwent PTH treatment (groups 6,7,8). The trabecular number (Tb.N) and trabecular separation (Tb.Sp) are similar in all treated groups (from 3 to 8); in particular, Tb.N values (from 3 to 8) are lower with respect to control groups (2, 9), while Tb.Sp values are higher than control groups. After 12 days, in all groups undergoing normal diet restoration (5, 6, 7, and 8) alizarin bone labeling shows that the trabecular mineral surface (Tb.MS/BS) is lower (sometimes with statistical significance) than recorded at the end of calcium-free diet feeding (groups 3 and 4). As far as L5 anterolateral cortical bone thickness (Ct.Th) is concerned (Figure 5), the value is reduced significantly after one month of calcium-free diet, independently of PTH treatment; otherwise, in all animals after normal diet restoration, the cortical bone thickness increases significantly, without reaching the value of the control animals. As far as L5 posterior cortical bone is concerned, both dietary regimen and PTH treatment do not affect the values of Ct.Th. The values of mineral apposition rate (MAR) do not show any relevant differences between all groups.

#### 2.2.2. Femoral Mid-Diaphysis

Static and dynamic histomorphometric parameters are reported in Figure 6: Ct.B.Ar values as well as periosteal MS and periosteal MAR do not show significant differences between all groups. Otherwise, endosteal MS is significantly higher (even double) in animals fed with a normal diet restoration (groups 5 to 8) with respect to animals fed a calcium-free diet for 1 month (groups 3 and 4); also, endosteal MAR is higher in animals fed with a normal diet restoration (5–8) with respect to animals fed a calcium-free diet (3 and 4), although without statistical significance. As shown in Figure 7, the presence of endosteal labels is mainly evident in normal diet restoration groups (from 5 to 8).

#### 2.2.3. Femoral Distal Metaphysis

Transverse femoral distal metaphysis sections, stained with Alizarin Red, used for static histomorphometry, are shown in Figure 8. Morphological analyses show that in all rats, apart from the ones in the control groups (groups 2 and 9), bony trabeculae appear rarefied in the posterior portion of femoral metaphyses. As shown in Figure 9, mean values of BV/TV, Tb.Th, and Tb.N increase, while those of Tb.Sp decrease (although without statistical significance) in animals fed with a normal diet restoration with respect to animals fed a calcium-free diet for one month (significant decrease only in group 7 vs. group 3 *p* < 0.05 of Tb.Sp values). After 12 days, in all groups undergoing normal diet restoration (5, 6, 7, and 8) alizarin bone labeling shows that the percentage of Tb.MS/BS is significantly lower than that recorded at the end of calcium-free diet feeding (groups 3 and 4). Figure 10 shows that new bone deposition is abundant (red fluorescence) mostly on the surface of the few trabecular remnants of groups 3 and 4. The values of Ct-B-Ar and cortical endosteal MAR do not show any relevant differences among all groups (Figure 11).

### 2.3. Sclerostin–Immunohistochemical Analysis

In transverse sections of L4, analysis of Sclerostin expression showed that the positivity of osteocytes located in the cortical bone of the anterolateral portion of the vertebral body is higher in animals fed a calcium-free diet (groups 3 and 4) with respect to all animals with normal diet restoration (groups 5–8) and control groups, independently from PTH (1-34) administration; the values (reported in Figure 12) are often statistically significant. 

### 2.4. Serum Biochemical Analysis

Table 2 reports the mean values of parameters from sera of all rats, collected at the end of the experimental period. The mean values of Ca, P, OPG, BALP, CrossLaps, and PTH (1-84) do not show significant differences among the various groups.

## 3. Discussion

For the exhaustive discussion of the present data, it is to be pointed out, as already above reported, that the observations described here must be interpreted together with those previously published [39].

Preliminarily, it is important to emphasize the differences between mineral and skeletal homeostasis: mineral homeostasis involves bone response to variation of dietary mineral content, while skeletal homeostasis implies bone answers to loading modifications (also depending on changes in weight).

In the previous study, it was demonstrated that the lack of calcium in the diet does not lead to only one type of bone response since, in the metabolic/biochemical osteoporosis, the interaction between mineral and skeletal homeostasis influences the pattern of bone loss in the different skeletal regions (appendicular vs. axial) with different metabolism-related architecture (trabecular vs. cortical bone). Based on the fact that PTH (1-34) seems to display good results as therapeutic support in recovering bone fragility and accelerating bone healing [29,31,40,41,42,43,44], one target of the present work is to determine whether, after one month of calcium-free feeding, normal diet restoration with/without concomitant PTH (1-34) administration can influence the pattern of bone mass recovery in terms of amounts and sites, not only depending on the mineral homeostasis, but also influenced by the skeletal homeostasis.

The first point to underline is the fact that, based on the skeletal homeostasis, the significant increase of body weight, observed during the entire experimental period with almost equal absolute values in all groups, can affect mostly those animals where the bone mass is greatly reduced as a consequence of the calcium-free diet, so that the non-resorbed bone is overloaded by weight. In turn, body weight affects the properties and the architecture of the skeleton just in response to the skeletal homeostasis [45].

After the expected reduction of trabecular bone of vertebral bodies and femoral metaphyses, due to the dietary imbalance, those bony trabeculae (not resorbed during the calcium free diet) thicken during normal diet restoration in consequence to new bone deposition since they are more devoted to answer to mechanical demand (i.e. complying the skeletal homeostasis). As far as the cortical bone is concerned, its answer to normal diet restoration is differently conditioned by the interaction with mechanical demands, depending on the skeletal regions involved: in the anterolateral portions of the vertebral body (that results overloaded due to the previous bone loss) the thickness significantly increases due to bone recovery at the endosteal side, while in the posterior parts of the vertebral body as well as in the femur metaphysis and mid-diaphysis (less involved in loading) it does not increase.

Regarding to PTH (1-34) administration, the results highlight its role in improving bone mass recovery only in specific skeletal sites; in fact, after two months, in rats that received PTH (1-34), indifferently during the calcium-free diet and/or during the normal diet restoration, the trabecular bone mass generally increases with respect to rats with normal diet restoration without PTH (1-34) administration (group 5) and with respect to animals sacrificed after one month of a calcium-free diet. In particular, at the vertebral level, the bone mass increment is mostly due to an increase (always significant) of thickness of the preexistent trabeculae rather than to the formation of new trabeculae; at the femoral level, trabecular bone shows a trend similar to the vertebral trabeculae during normal diet restoration. To explain this evidence, it is important to underline that the formation of new trabeculae or the thickening of preexisting trabeculae, depends respectively on two different types of osteogenesis, named static and dynamic [46]. In fact, as Ferretti and coworkers clearly demonstrated, static osteogenesis (SO) is conditioned by vascular-derived inductive stimuli that do not appear to affect our experimental evidences; on the contrary, dynamic osteogenesis (DO) is conditioned by mechanical stimuli (sensed by osteocytes), which is in turn dependent on the increase of the body weight occurring in our experimental condition. These results are also fully in line with the demonstration that the crucial effect of PTH (1-34) is to improve only the “dynamic” bone formation instead of the “static” one during bone repair in transcortical holes experimentally drilled in rat femur [47]. On the contrary, in some pathologies, like facet joint osteoarthritis (FJOA), remodeling of the subchondral trabecular bone compartment is characterized by increased trabecular number, rather than trabecular thickening [48]; this observation can be explained with the impairment of viability of osteocytes (i.e., the bone mechanosensor) inside the trabecular bone due to osteoarthritis, so that static osteogenesis (instead of dynamic one) is activated, through the recruitment of osteoprogenitor cells by endothelial-derived growth factors, giving rise to the formation of new trabeculae. Moreover, Kumabe and coworkers [43] showed that the administration of PTH (1–34) increases union rate and accelerates bone healing in rat refractory fracture models, suggesting that such a drug could become a useful therapy for accelerating fracture healing in patients at high risk of delayed union or nonunion. In our model, as far as cortical bone is concerned, at the mid-diaphyseal level no modifications were observed on bone mass in relation to PTH administration. The different effect of PTH (1-34) in trabecular bone (constituting the most part of the vertebral axial skeleton) versus cortical bone (constituting the most part of appendicular long bones) can be explained with the suggestion that PTH affects mostly those regions involved in mineral homeostasis (i.e., the vertebral bodies), which also interacts with skeletal homeostasis: in the first month (calcium-free diet) the number or vertebral trabeculae decreases in response to mineral homeostasis and in the second month (normal diet restoration) the remaining trabeculae increases in thickness for the increase of body weight.

The observations concerning the endosteal mineralizing surface (MS%) might seem apparently controversial, since the results showed that in all animals with normal diet restoration (with respect to rats fed calcium-free diet): (i) the cortical bone of femoral mid-diaphysis show higher values (often with statistical significance) and (ii) the trabecular bone of both vertebral body and femoral metaphysis show lower values (always with statistical significance). These observations are instead in line with what was mentioned above about the ability of trabecular bone to respond more readily to mineral homeostasis (in consequence of calcium depletion), which, in turn, led to higher bone loss; therefore, the few remaining trabeculae are overloaded and close to their surface new bone matrix is laid down (by means of dynamic osteogenesis) to answer to skeletal homeostasis. This occurs early, probably already towards the end of the calcium free diet: in fact, the trabeculae appear more labeled (Alizarin Red) with respect to rats with normal diet restoration. In line with the fact that trabecular architecture responds more promptly to both homeostases there is the observation that in femoral cortical bone the neo-osteogenesis is abundant later, during the normal diet restoration, as evidenced by bone labeling on the twelfth day. All this agrees with observations of various authors [49,50,51] showing that appendicular skeleton (mostly concerning cortical bone) answers mainly to mechanical demands (i.e., is devoted to the skeletal homeostasis) while axial skeleton (mostly concerning trabecular bone) answers mainly to metabolic demand (i.e., is mainly devoted to the mineral homeostasis).

As far as Sclerostin is concerned, it is not surprising that its expression is higher in animals fed calcium-free diet with respect to all animals with restored normal diet and to the control groups, since osteocytes are both the bone mechanosensor and the major producers of this protein inhibiting osteoblast activity, depending on the answers to mineral and skeletal homeostases [52]. Sclerostin expression seems not to be affected by PTH (1-34) administration.

Among all serum levels, it has to be noted that the mean values of Ca and P show similar values due to the good bone answer to both mineral and skeletal homeostases. We consider this aspect as a physiological compensatory mechanism to respond to the acute absence of calcium absorbed from diet, as also inferred by others [53,54,55]. The present results suggest that to compensate for a decreased serum calcium concentration the parathyroid hormone stimulated the excessive resorption of calcium from bone that, in turn, also releases phosphate into blood and, as consequence, serum phosphorus levels may increase. As far as OPG, BALP, CrossLaps, and PTH (1-84) are concerned, the individual variability among the animals inside the same group does not allow for discussion of the obtained data.

In conclusion, the main findings of the present study are (i) the importance of the interplay between mineral homeostasis and skeletal homeostasis in modulating and guiding bone answers to dietary/metabolic alterations and (ii) the evidence that the more involved bony architecture is the trabecular one, the most susceptible to the dynamical balance of the two homeostases. Clinical strategies in recovery of osteoporosis or any other skeletal impairment (of both metabolic or traumatic origin) must absolutely take into account mostly the type of the bony architecture involved, regardless of the bone mass to be recovered. Furthermore, it is also interesting to note that the main target of PTH (1-34) is the trabecular bone. Once again, therapeutic protocols should take into account these aspects to optimize the drug effect, thus speeding the recovery.

## 4. Materials and Methods

### 4.1. Experimental Animals and Treatment

Three-month-old Sprague Dawley male rats (*n* = 43) were purchased from Charles River Laboratories (Calco, Lecco, Italy). As previously reported [39], the choice of rat provides a good experimental animal model for studying bone remodeling alterations during biochemical osteoporosis. At the time of arrival (TA), all rats were housed individually in single cages, to better check food intake of each rat, and maintained under laboratory controlled conditions (22 ± 1 °C, 55–60% humidity, 12 h light:12 h dark). After seven days of acclimation to housing conditions (T0), the rats were randomized into nine groups (Figure 1), indicated as follows.

Group 1 (baseline, *n* = 3): sacrificed after 7 days of acclimation;Group 2 (control, *n* = 5): fed normal diet and natural water ad libitum for 4 weeks (T1);Group 3 (*n* = 5): fed calcium-deprived diet and distilled water ad libitum for 4 weeks (T1);Group 4 (*n* = 5): fed calcium-deprived diet and distilled water ad libitum for 4 weeks with concomitant administration of PTH (1-34) 40 μg/kg/day (T1);Group 5 (*n* = 5): fed calcium-deprived diet and distilled water ad libitum for 4 weeks (T1) and successive normal diet restoration and natural water ad libitum for 4 weeks (T2);Group 6 (*n* = 5): fed calcium-deprived diet and distilled water ad libitum for 4 weeks (T1) and successive normal diet restoration and natural water ad libitum for 4 weeks with concomitant administration of PTH (1-34) 40 μg/kg/day (T2);Group 7 (*n* = 5): fed calcium-deprived diet and distilled water ad libitum for 4 weeks with concomitant administration of PTH (1-34) 40 μg/kg/day (T1) and successive normal diet restoration and natural water ad libitum for 4 weeks (T2);Group 8 (*n* = 5): fed calcium-deprived diet and distilled water ad libitum for 4 weeks with concomitant administration of PTH (1-34) 40 μg/kg/day (T1) and successive normal diet restoration and natural water ad libitum for 4 weeks with concomitant administration of PTH (1-34) 40 μg/kg/day (T2);Group 9 (control, *n* = 5): fed normal diet and natural water ad libitum for 8 weeks (T2).

The amount of food intake by each rat was regularly checked, verifying the absence of significant differences in the feeding of each animal.

Groups 1, 2, 3, and 4 have been previously analyzed and the results were reported in the paper by Ferretti and coworkers [39]. The following descriptions refer to groups 5–9. The calcium-deprived diet is a casein-based synthetic diet containing a very low amount of calcium (0.04% Ca). PTH (1-34) was supplied by Eli Lilly and Company (Indianapolis, USA), solubilized in saline (40 µg/mL) and subcutaneously injected in a volume of 100 μL/100gr body weight per rat. During normal diet restoration, in order to evaluate the newly-formed bone, all animals underwent bone labeling subcutaneous injection: (i) first day—Calcein (Fluka, St. Louis, MO, USA) 15 mg/kg; (ii) sixth day—Oxytetracycline hydrochloride (Sigma, St. Louis, MO, USA) 30 mg/kg; and (iii) 12th day—Alizarin Red S (Fluka, St.Louis, MO, USA) 30 mg/kg. The choice of these bone labeling times was dictated by the need to observe the eventual bone deposition in the first two weeks after the restoration of the normal diet.

The body weight of each animal was recorded at the time of arrival (TA) in the housing facility, at the beginning of the experimental period (T0), after four weeks (T1), and after eight weeks (T2). At the end of the treatment, all rats were anesthetized with ether and blood samples were collected by cardiac puncture; then, rats were euthanized by exsanguination under ether anesthesia.

All experiments were carried out according to the Bioethical Committee of the Italian National Institute of Health. Animal care, maintenance, and surgery were conducted in accordance with Italian law (D.L. number 116/1992) and European legislation (EEC number 86/609). The experimentation protocol (n. 14-15012010 of Local Animal House-Ethical Committee was authorized (identification code DM n. 191/2010-B, approved on 20.10.2010) by Italian Ministry of Health.

### 4.2. Histology and Histomorphometry

Immediately after euthanasia, the fifth lumbar vertebra (L5) and the right femur of each animal were removed, deprived of soft tissues, fixed in sodium phosphate-buffered (PBS) 4% paraformaldehyde pH 7.4, dehydrated in graded ethanol, and embedded in methyl-methacrylate resin (Sigma Aldrich, Milan, Italy). Vertebrae and femurs were transversally cut with a Leica SP 1600 diamond saw microtome cutting system (Leica SpA, Milan, Italy) to obtain serial 200 μm thick sections. The sections, taken from the central level of the lumbar vertebral body and from the femur mid-diaphysis and distal metaphysis (in particular, the more proximal section of the patellar grove), were glued to a glass slide and ground to a final thickness of 40 μm. As previously reported [39], sections were superficially stained with Alizarin Red, to clearly observe the mineralized bone matrix, and scanned with Epson 3200 perfection scanner at 3200 dpi resolution to perform histomorphometry by means of the software Image J (NIH, Bethesda, USA).

“Static histomorphometric parameters” were calculated in different skeletal regions:In vertebral body and femoral distal metaphysis: trabecular bone volume (BV/TV), trabecular thickness (Tb.Th), trabecular number (Tb.N), and trabecular separation (Tb.Sp);In anterolateral and posterior sides of the vertebral body: cortical bone thickness (Ct.Th);In femoral mid-diaphysis and distal metaphysis: cortical bone area (Ct.B.Ar).

To evaluate the “dynamic histomorphometric parameters”, the same sections were also further polished to remove the Alizarin Red staining and used for the estimation of bone deposition, based on the labeling technique. Bone labeling allows to measure the newly-formed bone during the first two weeks of the normal diet restoration and to distinguish it with respect the preexistent bone. The sections were photographed using a Nikon Eclipse 90i microscope (Tokyo, Japan) equipped with a DS-Fi1 Nikon digital camera and driven by the Nikon ACT-2U software; dynamic histomorphometric parameters were evaluated by means of the image analysis system software Image J (NIH, Bethesda, Rockville, MD, USA). The mineral apposition rate (MAR) was measured between the first and third label (Calcein and Alizarin, respectively) on the periosteal anterior and endosteal posterior sides of the vertebral cortical bone as well as on the periosteal and endosteal sides of the femoral mid-diaphyseal cortical bone and the endosteal cortical side of femur metaphysis. Moreover, the mineralizing surface with respect to the bone total surface marked with Alizarin (MS/BS) was measured on the trabecular bone of both vertebral body and femoral metaphysis as well as on periosteal and endosteal sides of the femoral mid-diaphysis. All measurements were performed according to the ASBMR histomorphometry nomenclature [56].

### 4.3. Immunohistochemical Analysis

At sacrifice, the fourth lumbar vertebra (L4) of each animal was removed, deprived of soft tissues, fixed in sodium phosphate-buffered (PBS) 4% paraformaldehyde pH 7.4, and decalcified in 10% EDTA solution for 30 days. Bones were then washed with PBS, dehydrated in graded ethanol, and embedded in paraffin. Transverse 5 μm-thick sections were obtained in the central portions of the vertebral body with a Leica RM2255 microtome (Leica SpA, Milan, Italy). The tissue sections were permeabilized with 0.3% Triton X-100, blocked for intrinsic peroxidase activity with 3% hydrogen peroxide for 10 min, and processed for antigen retrieval with Tris-EDTA (10 mM Tris Base, 1 mM EDTA Solution, 0.05% Tween 20, pH 9.0). Then, the slides were blocked in 10% normal serum with 1% BSA in TBS for two hours at room temperature and incubated with primary antibody (Sclerostin ABCAM - ab 63097) at 4 °C for 16 h. For detection of Sclerostin, the sections were incubated with secondary antibody anti-rabbit-HRP (Thermo Scientific) and the binding of peroxidase-conjugated secondary antibodies was detected with a DAB kit (Vector Lab, Peterborough, United Kingdom). Three to five immunostained sections for each vertebra were analyzed quantitatively. Compositions of images (magnification ×20) were made acquiring multiple fields from the anterolateral portion of the L4 cortical bone; both DAB-stained and unstained lacunae were counted and recorded using Image J software. The number (N) of stained osteocytes was related to the area (mm^2^) examined in each microscopic field and is expressed as N/mm^2^.

### 4.4. Serum Biochemical Analysis

Blood samples were centrifuged to separate serum that was preserved in tubes, immediately separated by centrifugation (4 °C) at 1500*g* for 15 min. Sera were then aliquoted into small volumes and stored at −20 °C for successive analyses. The levels of total calcium (Ca) and inorganic phosphorus (P) in serum were determined using the high performance Beckman Coulter analyzer AU680 Chemistry System. Immunometric assays for the determination of levels of osteoprotegerin (OPG), specific bone alkaline phosphatase (BALP), CTX (Beta CrossLaps), and bioactive intact PTH (1-84) in rat serum were purchased by Pantec s.r.l. (Turin, Italy); all kits are intended for research use only. In particular, rat-OPG and rat-BALP are two ELISA kits produced by SunRed Hotecnology Company (Shanghai, China); RatLaps is an EIA kit produced by Immunodiagnostic Systems Ltd. (Boldon, UK); rat bioactive intact PTH is an ELISA method produced by Immunotopics Inc. (San Clemente, CA, USA). The small amounts of reagents supplied in the kits prevented the possibility to perform automated procedures on laboratory analytical platform. To minimize the variables influencing the test, all good laboratory practice principles were applied: immediate storage of samples after serum separation at −20 °C; manual execution of tests in close agreement to the manufacturer’s instructions; execution of all tests over two consecutive days to avoid repeated freeze/thaw cycles of samples.

### 4.5. Statistical Analysis

Body weight for each rat was compared at different times by means of the paired t test; body weight variations among groups at the same time points were evaluated by means of one-way analysis of variance (ANOVA) followed by Bonferroni’s post hoc test. Histomorphometric parameters, immunohistochemical analysis, and serum levels were compared among groups by means of one-way analysis of variance (ANOVA) with Bonferroni’s test. All statistical analyses were performed using the Software STATA 11.0 (StataCorp, TX, USA). Values of *p* < 0.05 indicate significant differences between groups.

## Figures and Tables

**Figure 1 ijms-20-00753-f001:**
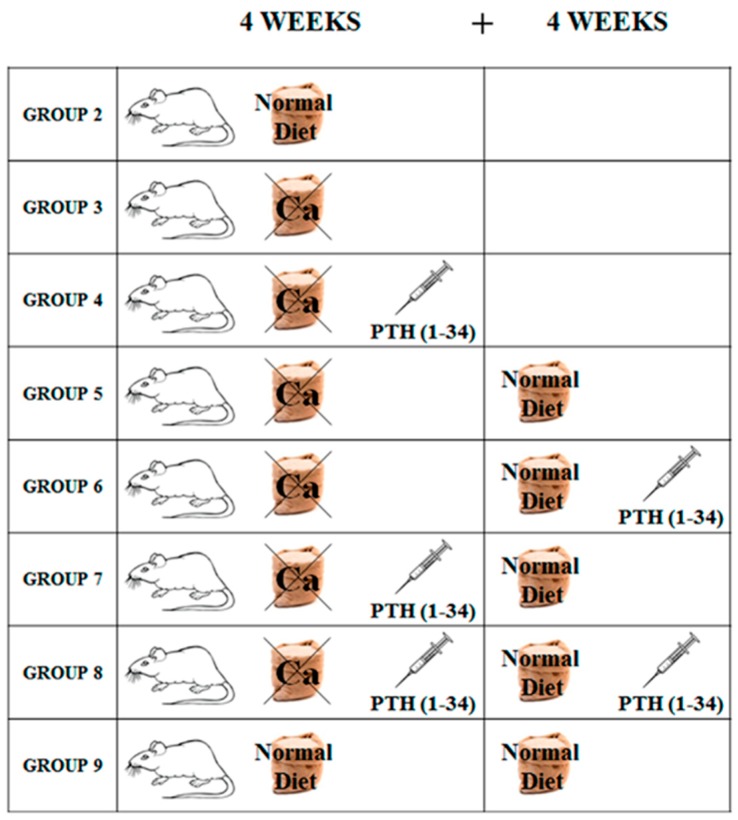
Schematic drawing showing the rats randomized in groups for the different experimental conditions. Group 2: control, normal diet for 4 weeks. Group 3: calcium-deprived diet for 4 weeks. Group 4: calcium-deprived diet with concomitant administration of PTH (1-34) 40 μg/kg/day for 4 weeks. Group 5: calcium-deprived diet for 4 weeks and successive normal diet restoration for 4 weeks. Group 6: calcium-deprived diet for 4 weeks and successive normal diet restoration with concomitant administration of PTH (1-34) 40 μg/kg/day for 4 weeks. Group 7: calcium-deprived diet with concomitant administration of PTH (1-34) 40 μg/kg/day for 4 weeks and successive normal diet restoration for 4 weeks. Group 8: calcium-deprived diet with concomitant administration of PTH (1-34) 40 μg/kg/day for four weeks and successive normal diet restoration with concomitant administration of PTH (1-34) 40 μg/kg/day for 4 weeks. Group 9: control, normal diet for eight weeks.

**Figure 2 ijms-20-00753-f002:**
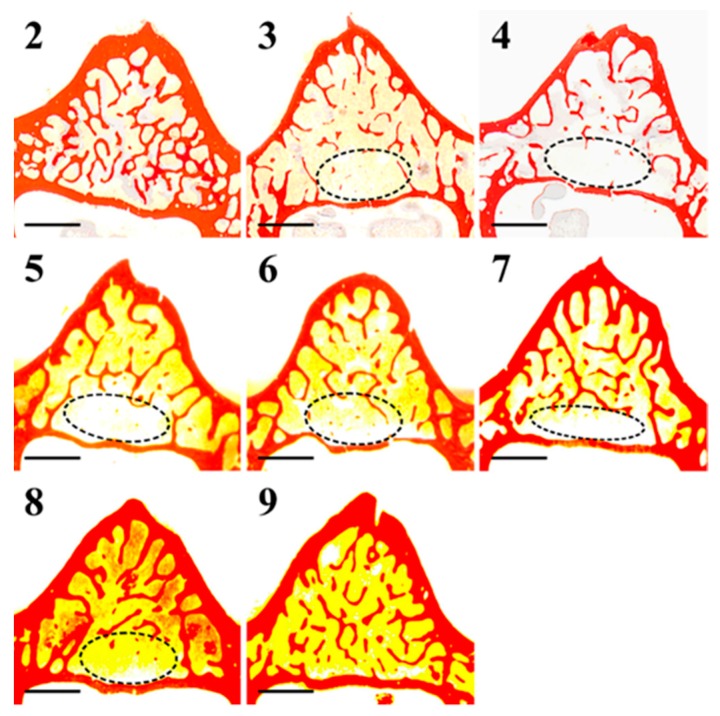
Scans of transversal sections of L5 body from all animal groups (2–9). The encircled areas show, in the posterior portion of the vertebral body, the almost-total absence of trabecular bone. Scale bar: 1 mm.

**Figure 3 ijms-20-00753-f003:**
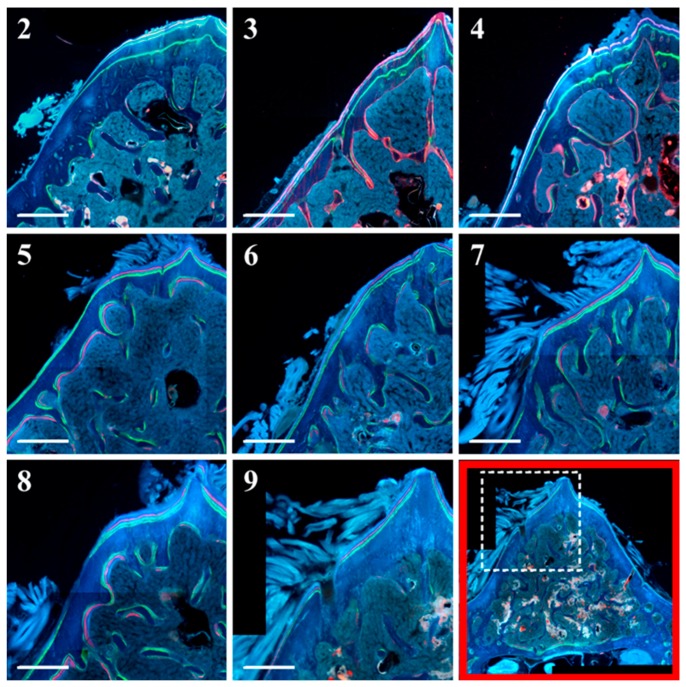
Micrographs of transverse sections of the anterolateral portion of L5 body from all animal groups under fluorescence microscope. In the bottom right image (outlined in red, as an example) the white dashed rectangle indicates the vertebral portion showed in all micrographs (2–9). Note on the surface of the few trabecular remnants (groups 3 and 4) the abundant red fluorescence. Note also, in the anterolateral cortex of the periosteal side (all groups), the presence of the three labels; in normal diet restoration only (groups from 5 to 8), the labels are well visible also on the endosteal side. Scale bar: 500 μm.

**Figure 4 ijms-20-00753-f004:**
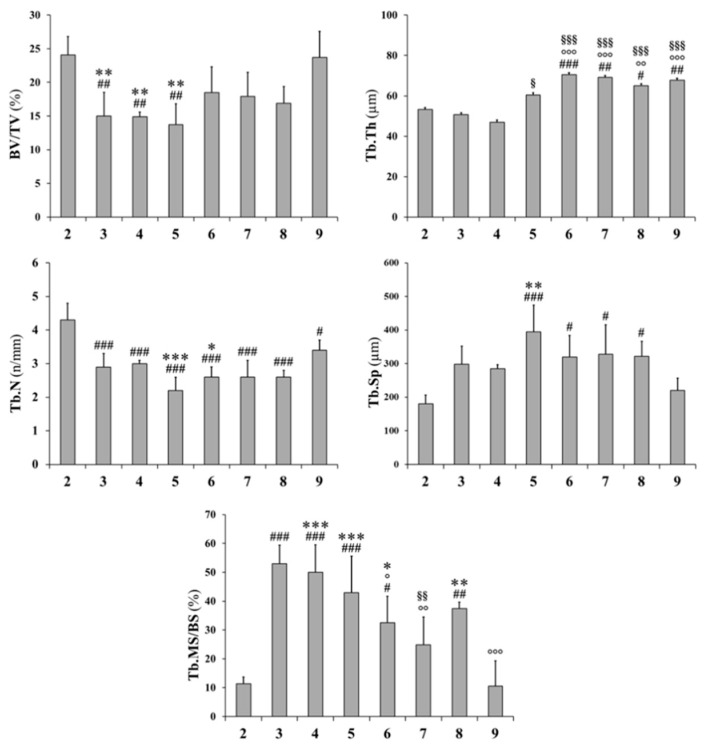
Static and dynamic histomorphometric parameters of trabecular bone in L5 vertebral body sections. BV/TV: trabecular bone volume; Tb.Th: trabecular thickness; Tb.N: trabecular number; Tb.Sp: trabecular separation; Tb.MS/BS: trabecular mineralizing surface. All values are expressed as mean ± S.D. Analysis of variance (ANOVA) followed by Bonferroni’s test: # *p* < 0.05, ## *p* < 0.01, ### *p* < 0.001 versus Group 2; ° *p* < 0.05, °° *p* < 0.01, °°° *p* < 0.001 versus Group 3; .§§ *p* < 0.01, §§§ *p* < 0.001 versus Group 4; * *p* < 0.05, ** *p* < 0.01, *** *p* < 0.001 versus Group 9.

**Figure 5 ijms-20-00753-f005:**
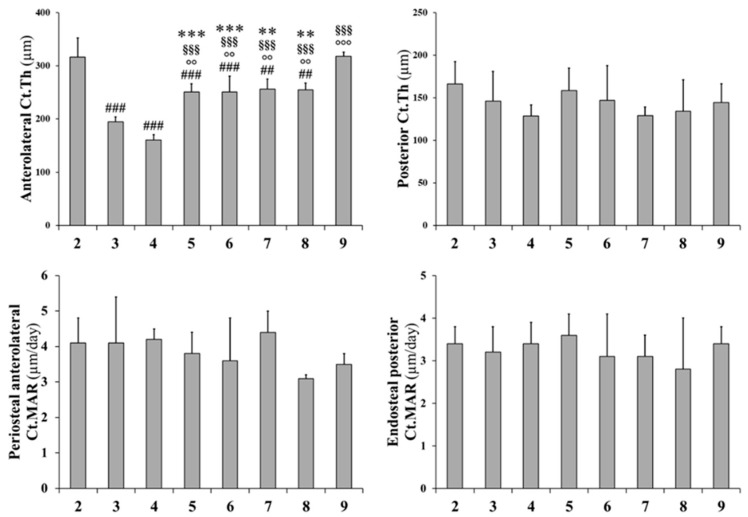
Static and dynamic histomorphometric parameters of cortical bone in L5 vertebral body sections. Ct.Th: cortical bone thickness; Ct.MAR: cortical mineral apposition rate. Values are expressed as mean ± S.D. ANOVA followed by Bonferroni’s test: ## *p* < 0.01, ### *p* < 0.001 versus Group 2; °° *p* < 0.01, °°° *p* < 0.001 versus Group 3; §§§ p < 0.001 versus Group 4; ** *p* < 0.01, *** *p* < 0.001 versus Group 9.

**Figure 6 ijms-20-00753-f006:**
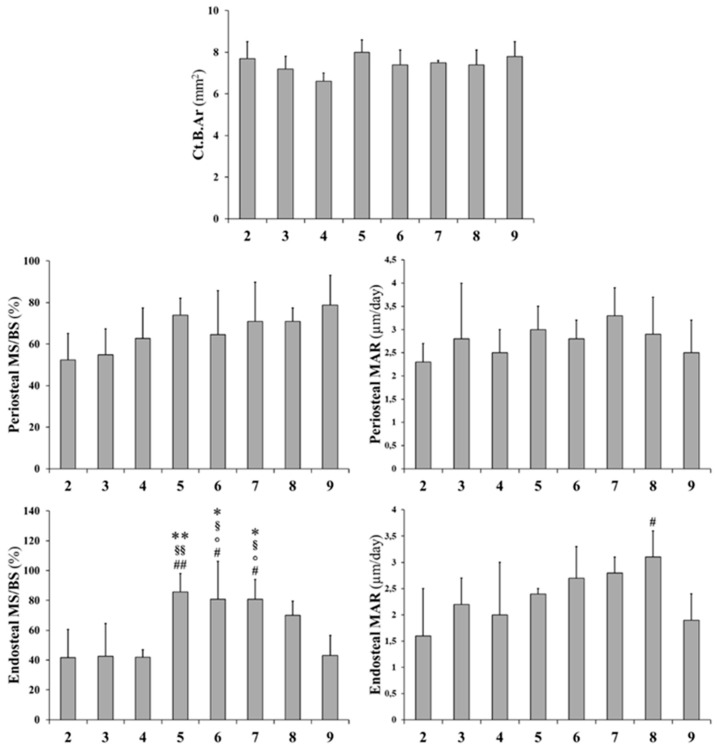
Static and dynamic histomorphometric parameters of cortical bone in mid-diaphyseal femoral sections. Ct.B.Ar: cortical bone area; MS/BS: mineralizing surface; MAR: mineral apposition rate. Values are expressed as mean ± S.D. ANOVA followed by Bonferroni’s test. # *p* < 0.05, ## *p* < 0.01 versus Group 2; ° *p* < 0.05 versus Group 3; § *p* < 0.05, §§ *p* < 0.01 versus Group 4; * *p* < 0.05, ** *p* < 0.01 versus Group 9.

**Figure 7 ijms-20-00753-f007:**
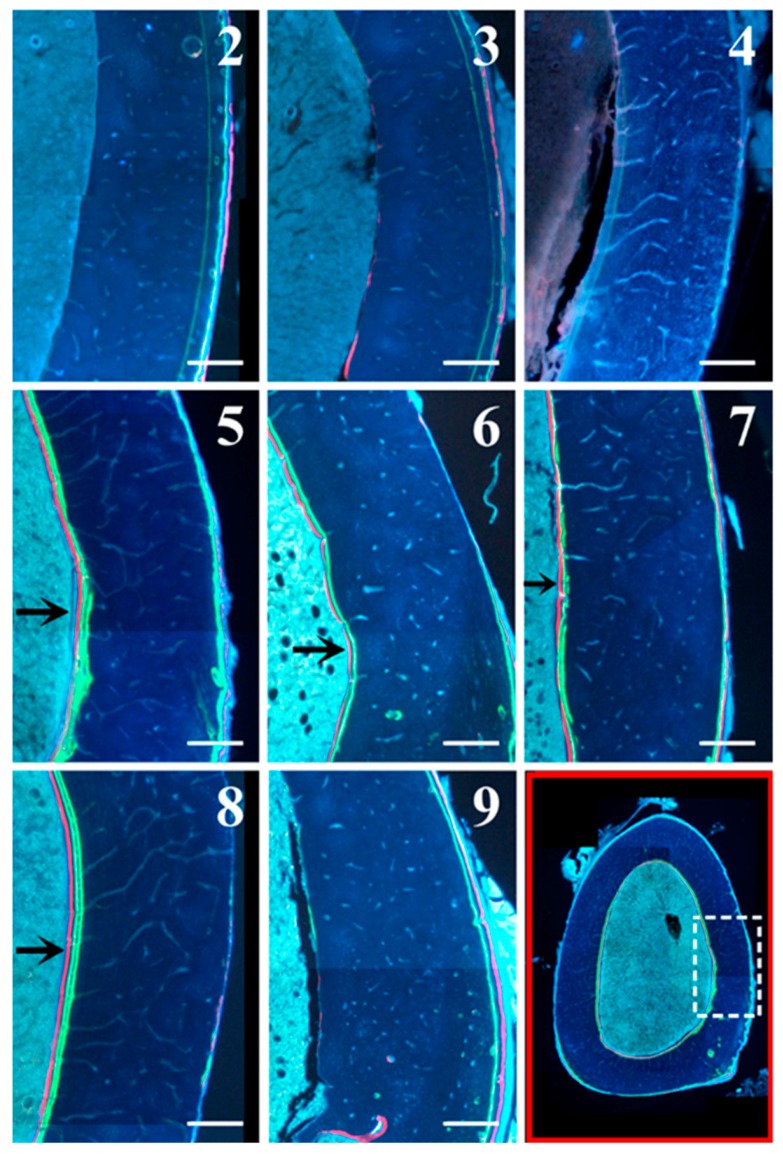
Details of femoral mid-diaphysis transversal sections of all animal groups under fluorescence microscope. In the bottom right image (outlined in red, as an example) the white dashed rectangle indicates the femur diaphysis portion showed in all micrographs (2–9). Note the presence of endosteal labels (arrows) in normal diet restoration groups (from 5 to 8). Scale bar: 250 μm.

**Figure 8 ijms-20-00753-f008:**
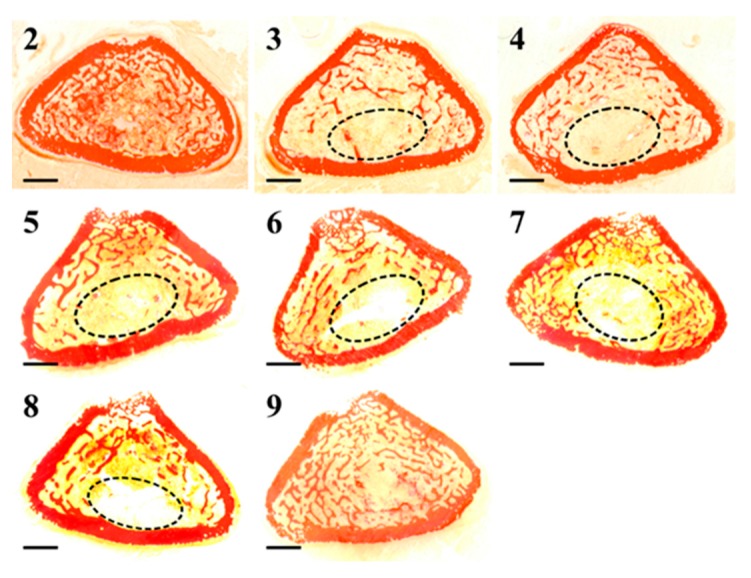
Scans of femoral distal metaphysis transversal sections of all animal groups (2–9). The encircled areas show rarefied bony trabeculae in the posterior portion of the femoral metaphyses. Scale bar: 1 mm.

**Figure 9 ijms-20-00753-f009:**
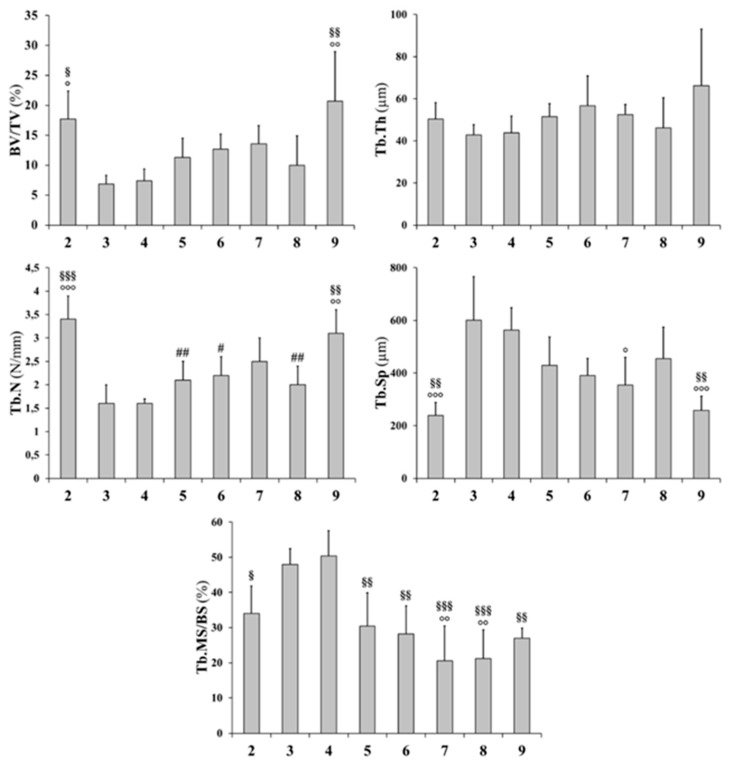
Static and dynamic histomorphometric parameters of trabecular bone in femoral distal metaphyseal sections. BV/TV: trabecular bone volume; Tb.Th: trabecular thickness; Tb.N: trabecular number; Tb.Sp: trabecular separation; Tb.MS/BS: trabecular mineralizing surface. Values are expressed as mean ± S.D. ANOVA followed by Bonferroni’s test. # *p* < 0.05, ## *p* < 0.01 versus Group 2; ° *p* < 0.05, °° *p* < 0.01, °°° *p* < 0.001 versus Group 3; § *p* < 0.05, §§ *p* < 0.01, §§§ *p* < 0.001 versus Group 4.

**Figure 10 ijms-20-00753-f010:**
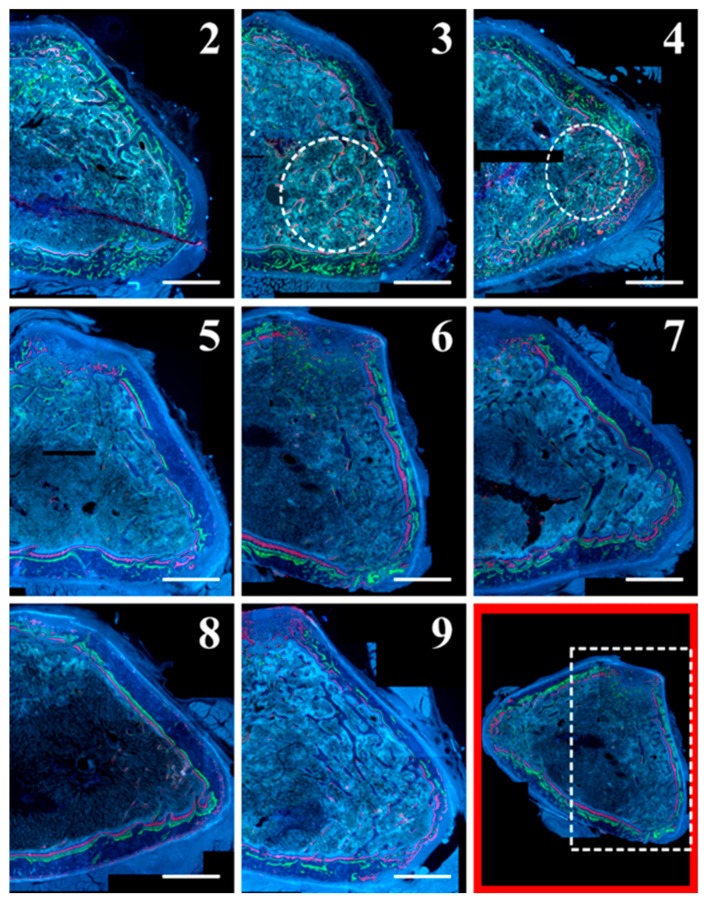
Details of micrographs from femoral distal metaphysis transversal sections of all animal groups under fluorescence microscope. In the bottom right image outlined in red (as an example), the white dashed rectangle indicates the femur metaphysis portion showed in all micrographs (2–9). Note the abundant red fluorescence mostly on the surface of the few trabecular remnants of groups 3 and 4 (encircled areas). Scale bar: 1 mm.

**Figure 11 ijms-20-00753-f011:**
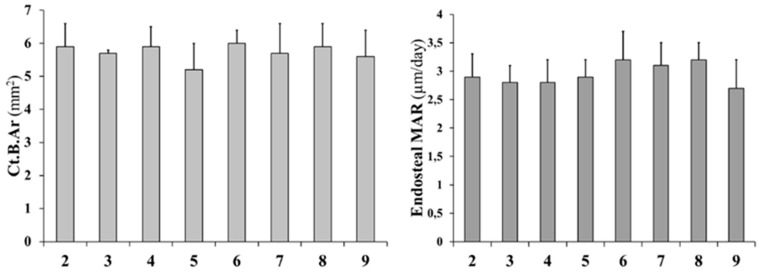
Static and dynamic histomorphometric parameters of cortical bone in femoral distal metaphyseal sections. Ct.B.Ar: cortical bone area; MAR: mineral apposition rate. Values are expressed as mean ± S.D. ANOVA followed by Bonferroni’s test.

**Figure 12 ijms-20-00753-f012:**
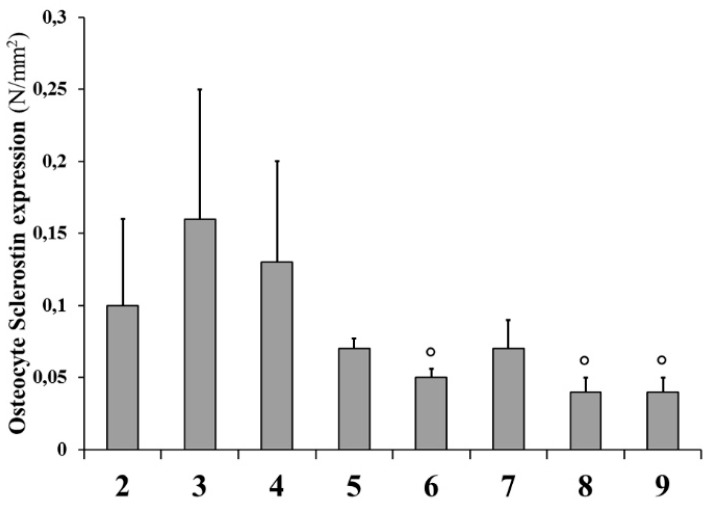
Sclerostin expression of osteocytes in the cortical bone of the anterolateral portion of the L4 vertebral body. The number (N) of stained osteocytes was related to the area (mm^2^) examined in each microscopic field. Values are expressed as mean ± S.D. ANOVA followed by Bonferroni’s test. ° *p* < 0.05 versus Group 3.

**Table 1 ijms-20-00753-t001:** Body weights of rats at the time of arrival (TA), at the beginning (T0), after four weeks (T1), and after eight weeks (T2) of experimentation.

Group	Weight at TA	Weight at T0	Weight at T1	Weight at T2
2	405 ± 32	436 ± 35 °°°	526 ± 47 ***	
3	405 ± 32	429 ± 37 °°	528 ± 70 **	
4	397 ± 25	422 ± 21 °°	504 ± 40 **	
5	398 ± 24	423 ± 30 °°°	505 ± 48 **	551 ± 65 ^##^
6	400 ± 26	417 ± 23 °°	518 ± 28 **	533 ± 30
7	403 ± 29	425 ± 29 °°	507 ± 34 ***	555 ± 44 ^##^
8	405 ± 28	428 ± 36 °°	503 ± 55 **	534 ± 62 ^##^
9	406 ± 28	430 ± 29 °°°	523 ± 54 **	559 ± 74 ^#^

All values are expressed as mean ± S.D. Paired *t-*test. °° *p* < 0.01, °°° *p* < 0.001 versus weight at TA; ** *p* < 0.01, *** *p* < 0.001 versus weight at T0; # *p* < 0.05, ## *p* < 0.01 versus weight at T1.

**Table 2 ijms-20-00753-t002:** Mean values of serum levels at the end of each trial period: Ca, P, OPG, BALP, CrossLaps, and PTH (1-84).

Group	Ca mg/dL	P mg/dL	OPG ng/mL	BALP ng/mL	Cross Laps ng/mL	PTH (1-84) pg/mL
2	9.74 ± 0.28	6.65 ± 0.92	0.71 ± 0.04	6.90 ± 0.63	51.91 ± 19.16	54.52 ± 25.85
3	10.01 ± 0.16	7.35 ± 0.87	0.67 ± 0.10	6.84 ± 0.41	66.36 ± 17.20	78.88 ± 29.79
4	10.12 ± 0.28	7.04 ± 0.72	0.69 ± 0.05	6.63 ± 0.37	58.24 ± 20.88	62.41 ± 23.35
5	10.15 ± 0.27	5.89 ± 0.41	0.69 ± 0.07	6.51 ± 1.32	44.84 ± 11.86	58.41 ± 12.65
6	10.34 ± 0.24	6.10 ± 0.86	0.74 ± 0.10	6.97 ± 0.41	35.07 ± 4.55	80.76 ± 40.22
7	10.20 ± 0.16	6.18 ± 0.67	0.65 ± 0.06	6.41 ± 0.73	40.38 ± 22.07	112.99 ± 79.19
8	10.15 ± 0.19	6.12 ± 0.43	0.69 ± 0.04	6.69 ± 0.55	58.68 ± 42.44	72.05 ± 49.77
9	10.25 ± 0.30	7.10 ± 1.09	0.65 ± 0.03	6.07 ± 1.5	48.89 ± 15.66	63.35 ± 38.47

All values are expressed as mean ± S.D.

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
