# Peer review of "Interaction among Calcium Diet Content, PTH (1-34) Treatment and Balance of Bone Homeostasis in Rat Model: The Trabecular Bone as Keystone"

_ijms, 2019, doi:10.3390/ijms20030753_

Reviewer 1 Report

This is a very comprehensive analysis of the effect of normal-diet restoration,

and PTH(1-34) treatment, on bone mass. The study has a wealth of high quality data (histology, histomorphometry, serum biomarker data, etc). The results showed that trabecular bone responds to changes in calcium diet content to a larger extent than cortical bone and that PTH has a large effect in the recovery of trabecular bone. These results could have a clinical impact in the treatment of osteoporosis. I recommend to send it to an Editor for English language grammar.

Author Response

The authors thank very much the Reviewer for the appreciation of the work done.

Editing of English language and style was made by an accredited native speaker, cited in the Acknowledgments.

Reviewer 2 Report

The authors describe "Interplay among calcium diet content, PTH(1-34) treatment and balance of bone homeostases in rat model: the trabecular bone as keystone."

The present study is the second step (concerning the normal-diet restoration) of the previous one (concerning the calcium-free diet) to determine whether the normal-diet restoration, with/without concomitant PTH(1-34) administration, can influence amounts and deposition sites of the total bone mass. Histomorphometric evaluations and immunohistochemical analysis for Sclerostin expression were conducted on the vertebral bodies and femurs in rat model. The final goals are: i) to define timing and manners of bone mass changes when calcium is restored in the diet; ii) to analyze the different involvement of the two bony architectures having different metabolism (i.e. trabecular versus cortical bone); iii) to verify the eventual role of PTH(1-34) administration. Results evidenced the greater involvement of the trabecular bone with respect to the cortical one, in answering to different calcium diet content, and the effect of PTH mostly in the recovery of trabecular bony architecture. The main findings emerged from the present study are: i) the importance of the interplay between mineral homeostasis and skeletal homeostasis in modulating and guiding bone answers to dietary/metabolic alterations and ii) the evidence that the more involved bony architecture is the trabecular one, the most susceptible to the dynamical balance of the two homeostases.

Although the manuscript is written concisely and will be very interesting, several concerns are raised.

Is there any model mouse with deficiency of PTH?

Current model is not good for the puropose of the proposed experiment.

Since it is normal mouse, under the condition of calcium free, PTH level has been increased as expected. Other group of mice are as well. In that sense, not much novel finding.

How about other bone markers including C-terminal and N-terminal pro-peptide of procollagen type I (PINP), Type IIA collagen N-Propeptide (PIIANP), collagen X, C-type natriuretic peptide (CNP), deoxypyridinoline (DPD), pyridinoline (PYD), calcitonin, N-terminal telopeptide of type 1 collagen (NTX) etc?

How about impact on growth plate at the knee joint and its extracellular matrix?

How about the size of chondrocytes?

Overall, the current version of the manuscript includes a very interesting point in funcion of PTH but some revisions with novel implications are required.

Author Respons

The authors thank the Reviewer for the suggestions.

Editing of English language and style was made by an accredited native speaker, cited in the Acknowledgments.

Raised points:

Reviewer: Is there any model mouse with deficiency of PTH?

Current model is not good for the puropose of the proposed experiment.

Authors: The authors thank the reviewer for the suggestion (PTH-deficiency mouse model) but underline that the choice of the animal experimental model was made at the time of the first work (Ferretti et al., 2015) [this aspect was underlined in the new version of M&M (lines 356-358)], in which a series of references (see below) was reported to justify the option selected (i.e. rat fed with a calcium-deprived diet instead of mouse model with primary parathyroidism, in which not-physiological conditions could interfere with the normal relationships between the two homeostases - mineral and skeletal).

M. Harrison and R. Fraser, “Bone structure and metabolism in calcium-deficient rats,” Journal of Endocrinology, vol. 21, pp.197–205, 1960.

J. Gershon-Cohen, J. F.McClendon, J. Jowsey, andW. C. Foster, “Osteoporosis produced and cured in rats by low and high calcium diets,” Radiology, vol. 78, no. 2, pp. 251–252, 1962.

C. D. Salomon, “Osteoporosis following calcium deficiency in rats,” Calcified Tissue Research, vol. 8, no. 1, pp. 320–333, 1971.

F. R. de Winter and R. Steendijk, “The effect of a low calcium diet in lactating rats; observations on the rapid development and repair of osteoporosis,” Calcified Tissue International, vol. 17, no. 4, pp. 303–316, 1975.

P. Rasmussen, “Calcium deficiency, pregnancy, and lactation in rats. Some effects on blood chemistry and the skeleton,” Calcified Tissue International, vol. 23, no. 1, pp. 87–94, 1977.

P. Rasmussen, “Calcium deficiency, pregnancy, and lactation in rats. Microscopic and microradiographic observations on bones,” Calcified Tissue International, vol. 23, no. 1, pp. 95–102, 1977.

H. A. Sissons, G. J. Kelman, and G. Marotti, “Mechanisms of bone resorption in calcium-deficient rats,” Calcified Tissue International, vol. 36, no. 6, pp. 711–721, 1984.

H. A. Sissons, G. J. Kelman, and G. Marotti, “Bone resorption in calcium-deficient rats,” Bone, vol. 6, no. 5, pp. 345–347, 1985.

U. Agata, J.-H. Park, S. Hattori et al., “The effect of different amounts of calcium intake on bone metabolism and arterial calcification in ovariectomized rats,” Journal of Nutritional Science and Vitaminology, vol. 59, no. 1, pp. 29–36, 2013.

V. Shen, R. Birchman, R. Xu, R. Lindsay, and D.W. Dempster, “Short-term changes in histomorphometric and biochemical turnover markers and bone mineral density in estrogen and/or dietary calcium-deficient rats,” Bone, vol. 16, no. 1, pp. 149–156, 1995.

R. S. Mazzeo, H. J. Donahue, and S. M. Horvath, “Endurance training and bone loss in calcium-deficient and ovariectomized rats,” Metabolism, vol. 37, no. 8, pp. 741–744, 1988.

H. J. Donahue, R. S. Mazzeo, and S. M. Horvath, “Endurance training and bone loss in calcium-deficient and ovariectomized rats,” Metabolism, vol. 37, no. 8, pp. 741–744, 1988.

T. Hara, T. Sato, M. Oka, S. Mori, and H. Shirai, “Effects of ovariectomy and/or dietary calcium deficiency on bone dynamics in the rat hard palate, mandible and proximal tibia,”Archives of Oral Biology, vol. 46, no. 5, pp. 443–451, 2001.

A.Hodgkinson, J. E. Aaron, A.Horsman,M. S.McLachlan, and B. E. Nrodin, “Effect of oophorectomy and calcium deprivation on bone mass in the rat,” Clinical Science and Molecular Medicine, vol. 54, no. 4, pp. 439–446, 1978.

Reviewer: Since it is normal mouse, under the condition of calcium free, PTH level has been increased as expected. Other group of mice are as well. In that sense, not much novel finding.

Authors: The novelties in our research are: a) the different answer of trabecular bone with respect the compact one, not only under mechanical variations (skeletal homeostasis) but also under metabolic unbalance (mineral homeostasis); b) the importance of the interplay between mineral homeostasis and skeletal homeostasis in modulating and guiding bone answers to dietary/metabolic alterations under which, again,  the more involved bony architecture is the trabecular one, the most susceptible to the dynamical balance of the two homeostases.

Reviewer: How about other bone markers including C-terminal and N-terminal pro-peptide of procollagen type I (PINP), Type IIA collagen N-Propeptide (PIIANP), collagen X, C-type natriuretic peptide(CNP), deoxypyridinoline (DPD), pyridinoline (PYD), calcitonin, N-terminal telopeptide of type 1 collagen (NTX) etc?

Authors: Excellent suggestion for the work we are planning. Thanks a lot.

Reviewer: How about impact on growth plate at the knee joint and its extracellular matrix? How about the size of chondrocytes?

Authors: We have not yet dealt directly with the interference of calcium deficiency on the metaphyseal plate of growing animals. Probably in the near future we will verify this aspect in the growing animal rather than in the adult animal.

Reviewer: Overall, the current version of the manuscript includes a very interesting point in function of PTH but some revisions with novel implications are required.

Authors: For novel implications, sophisticated bio-molecular analyses are necessary, that imply the use of  further animals. Works in progress are already planned in our Lab.

Round  2

Reviewer 2 Report

I have checked it. I am satisfied with the current version. Acceptable.